# Membrane Protein Structure Determination and Characterisation by Solution and Solid-State NMR

**DOI:** 10.3390/biology9110396

**Published:** 2020-11-12

**Authors:** Vivien Yeh, Alice Goode, Boyan B. Bonev

**Affiliations:** School of Life Sciences, University of Nottingham, Nottingham NG7 2UH, UK; vivien.yeh@nottingham.ac.uk (V.Y.); alice.goode@nottingham.ac.uk (A.G.)

**Keywords:** membrane protein, solid-state NMR, solution NMR, dynamic nuclear polarisation

## Abstract

**Simple Summary:**

Cells, life’s smallest units, are defined within the enclosure of thin, continuous membranes, which confine the molecular machinery required for the life and replication of cells. Crucially, membranes of cells establish and actively maintain distinctly different environments inside cells, including electrical and solute gradients vital to normal cellular functions. Membrane proteins are in charge of transport, electrical polarisation, signalling, membrane remodelling and other important functions. As such, membrane proteins are key drug targets and understanding their structure and function is essential to drug development and cellular control. Membrane proteins have physical characteristics that make such studies very challenging. Nuclear magnetic resonance is one advanced tool that enables structural studies of membrane proteins and their interactions at the atomic level of detail. We discuss the applications of NMR in solution and solid state to membrane protein studies alongside new developments in signal and sensitivity enhancement through dynamic nuclear polarisation.

**Abstract:**

Biological membranes define the interface of life and its basic unit, the cell. Membrane proteins play key roles in membrane functions, yet their structure and mechanisms remain poorly understood. Breakthroughs in crystallography and electron microscopy have invigorated structural analysis while failing to characterise key functional interactions with lipids, small molecules and membrane modulators, as well as their conformational polymorphism and dynamics. NMR is uniquely suited to resolving atomic environments within complex molecular assemblies and reporting on membrane organisation, protein structure, lipid and polysaccharide composition, conformational variations and molecular interactions. The main challenge in membrane protein studies at the atomic level remains the need for a membrane environment to support their fold. NMR studies in membrane mimetics and membranes of increasing complexity offer close to native environments for structural and molecular studies of membrane proteins. Solution NMR inherits high resolution from small molecule analysis, providing insights from detergent solubilised proteins and small molecular assemblies. Solid-state NMR achieves high resolution in membrane samples through fast sample spinning or sample alignment. Recent developments in dynamic nuclear polarisation NMR allow signal enhancement by orders of magnitude opening new opportunities for expanding the applications of NMR to studies of native membranes and whole cells.

## 1. Introduction

The plasma membrane separates and confines the cell from its environment and encapsulates this smallest and complete unit of life. The membrane consists largely of phospholipids and membrane proteins, which include peripheral proteins, attached or tethered to the membrane surfaces, as well as integral proteins that are partially or fully embedded into the membrane matrix and traverse one or both leaflets. It is estimated that around 30–40% of proteins in most genomes encode membrane proteins [1] which perform essential cellular functions as channels, transports and receptors and mediate signalling vital to cellular survival [2]. Around 60% of commercially exploited drug targets are membrane proteins [3], which shows their importance beyond their physiological role in the cell. Therefore, understanding the three-dimensional structure of membrane proteins is of enormous interest both to the biological research community and to healthcare and industry. Akin to soluble proteins, membrane protein structure determination is amenable to all main structural approaches, yet, uniquely, sample preparation remains a major challenge. Structure determination is limited in applications to membrane protein studies by the requirement for a hydrophobic membrane to support and maintain the folded functional state. X-ray crystallography remains the leading tool for elucidating the structure of membrane proteins with major recent contributions from cryo-TEM. Nuclear magnetic resonance (NMR) spectroscopy offers an alternative or complementary approach, which obviates the need for protein crystallisation and retains molecular flexibility and dynamics during the analysis, as well as maintaining the functional state of the protein. NMR also offers sub-atomic resolution in describing conformational variations of ligands and lipids bound to the membrane proteins and uniquely provides structural information from whole cells.

There are two main types of membrane protein: integral and peripheral membrane protein. Integral membrane proteins are permanently embedded in the membrane, either via transmembrane domain (TMD) topology or tethered by a membrane anchor. Peripheral membrane proteins may associate with the membrane temporarily via electrostatic or other interactions while coexisting with a soluble fraction in the surrounding aqueous environment [4]. Almost all integral membrane proteins fold following a transmembrane topology that belongs to one of two major classes: α-helical bundles and β-barrels. Transmembrane proteins with α helical organisation have higher flexibility to accommodate conformational changes, often functioning as receptors and channels. G-protein couple receptors (GPCR), one of the largest membrane protein families in eukaryotes, consist of seven transmembrane helices and a short surface-localised helix [5,6]. While α-helical membrane proteins are common in plasma membranes, membrane proteins with β-barrel TMDs are only reported in outer membranes of Gram-negative bacteria, mitochondria and chloroplasts [7]. Transmembrane β-barrels function as selective or non-selective pores, transporters and secretory complexes, or can be involved in outer membrane biosynthesis and assembly [8].

## 2. Nuclear Magnetic Resonance

NMR spectroscopy relies on monitoring nuclear spin characteristics and dynamics to determine structures and interactions within and between molecules by discerning subtle changes in the local chemical atomic environment. NMR spectroscopy can be broadly categorised as solution NMR or solid-state NMR. While both are based on the same quantum mechanical principles of observing nuclear spin oscillations, they have different applications, advantages, limitations and experimental implementations and follow different methods of sample preparation. To illustrate the latter point, one may consider a simple cellular lysate centrifugation, after which proteins in solution would be suitable for solution NMR studies, while membrane pellets would be best characterised by solid-state NMR. Because of the marked differences in sample properties, solution and solid-state NMR have been used to address different and specific questions and can often be used in combination to overcome the limitations of using each method alone. Uniquely, NMR is a versatile tool in studying protein–ligand interaction, as ligand atoms are observed directly and conformational rearrangements within ligands on binding are reported in a sensitive manner through a variety of NMR characteristics including chemical shifts, nuclear relaxation and dynamics [9]. Chemical shift perturbation and changes in linewidth are some of the commonly used techniques in studies of protein–ligand interaction and can serve as tools for ligand screening and drug discovery [10]. Both solution NMR and solid-state NMR can be used to characterise protein–protein interactions and complex structures, using a variety of techniques such as chemical shift perturbation, solvent paramagnetic relaxation enhancement and heteronuclear dipolar recoupling [11].

Solution NMR methods are widely applied to structural and interaction studies of proteins, typically interrogating protein structure via probing backbone and amino acid sidechain conformations and interactions to derive distance and torsion angle constraints. These techniques can also be transferred to studies of membrane proteins prepared in suitable solubilising membrane mimetics. However, the spectral resolution in solution NMR studies depends strongly on molecular reorientation and decreases reciprocally with protein or proteomimetic volume. As a consequence, the application of solution NMR is limited by molecular size, which limits the range of proteins or complexes that can be studied. Recent technological advances in high-field magnets and cryogenic probes, combined with novel sample preparation protocols and transverse relaxation-optimised methods [12], have pushed the solution NMR protein size boundaries to nearly 100 kDa in some exceptional cases. In some applications, solution NMR has been used successfully to study membrane protein folding, interactions, conformational changes and internal mobility, as well as interaction with ligands and substrates.

Solid-state NMR is a versatile technique, which can be applied to protein samples in virtually any state but is developed to analyse insoluble systems, such as membrane proteins and biopolymers. Solid-state NMR is uniquely applicable to studies of membrane proteins within lipid bilayers, particularly if the structure, conformation and function are modulated by the composition of the membrane environment. Unlike solution NMR, the applications of solid-state NMR are not limited by the size of the investigated molecular complexes, nor by the viscosity of the medium, as achieving high resolution does not rely on thermal molecular reorientation in solution. Experimental challenges and resolution limitations in the solid state are primarily the result of anisotropic components of the nuclear interactions, sample heterogeneity or can be limited in large protein systems by spectral complexity and signal degeneracy, especially in sequential repeats of the same residue. Despite these limitations, solid-state NMR has been used successfully for determination of membrane protein structure in lipid bilayers, changes in protein conformation dependent on lipid composition and of protein orientation in the membrane.

The observation of NMR and its application to protein studies are uniquely conditional on the abundance of magnetically active nuclei with non-zero nuclear spin. As such, not all biologically relevant nuclei can be observed and used as NMR reporters. Different isotopes of each element possess different spin quantum numbers and gyromagnetic ratios, γ, and display unique NMR characteristics. The most abundant and commonly used nuclear system is protium, ^1^H or simply proton. Protons have spin I = 1/2, which results in simple spectral characteristics; protium is the most abundant atom type in biological macromolecules, with good representation in different chemical environments, and has high γ, which results in good signal intensity. As a result, protons are the most commonly used reporters in solution NMR studies and are also extensively used in solid-state NMR.

The high prevalence of ^1^H and the existence of extensive interproton networks in proteins and other biological macromolecules provide a wealth of structural information, yet these very same properties can also lead to spectral overcrowding and reduced resolution that results from strong residual interproton magnetic couplings. One approach used to modulate interproton couplings is isotope dilution, in which random or strategic substitution of protons for deuterium, ^2^H, can simplify the NMR spectra, reduce couplings and improve resolution, which is particularly useful for larger membrane proteins or protein complexes. Partial deuteration can be achieved in recombinantly produced proteins by growing the expression system on controlled media hydrated in deuterium oxide, or by addition of deuterated amino acid supplements and carbon sources. Deuteration can also enhance the NMR signal in observation of other nuclear systems, as ^2^H is 15 times more effective than ^1^H as a relaxation sink. Overall, replacing ^1^H with ^2^H would improve sensitivity and spectral resolution, which, implemented to the right measure, will also retain structural information encoded in the proton dipolar networks [13].

The second most abundant nuclear system in biological macromolecules is carbon, which, alongside nitrogen and oxygen, comprises almost all elements that participate in protein structures. While ^12^C and ^16^O are spinless and ^14^N is a quadrupolar system with complex spectroscopic behaviour, the stable isotopes ^13^C and ^15^N, naturally present at 1.1% and 0.4% in biological carbon and nitrogen, are excellent spin I = 1/2 NMR reporters and often used in NMR studies of membrane proteins alone or in combination with ^1^H and each other. Carbon-13 is particularly suitable due to its high chemical shift dispersion (*ca.* 200 ppm for proteins) and can be used in its natural abundance within biological macromolecules or after isotope substitutions from ^13^C-enriched samples. Natural abundance ^13^C spectroscopy is characterised by very high resolution from an isotopically diluted nuclear system but also is limited by its low signal intensity and vanishing ^13^C-^13^C correlation networks. Nitrogen-15 is rarely used alone but most commonly in combination with ^1^H, ^13^C or both in probing protein backbone structure or amide bond orientation in aligned membranes. With ^15^N only present naturally at 0.4% in proteins and due to its low γ, it is almost universal practice to enrich isotopically recombinant proteins via host strain growth on isotope-enriched or designer media. Commonly used carbon and nitrogen-limited uniform enrichment sources include ^13^C-glucose and ^15^N-NH_4_Cl for *E. coli*-based production in minimal media, although pyruvate, succinate or other carbon sources can be used instead. Other methods may include providing isotope-containing nutrients during protein expression, such as labelled amino acids [14]. Stable isotope substitutions are isomorphic and do not alter the chemical properties of atoms, protein structure, folding or interactions.

Uniformly ^13^C-labelled samples offer a wealth of structural information as torsion angle and distance restraints, which can be interrogated by ^13^C-^13^C correlation methods, particularly in solid-state NMR studies of membrane proteins. Common in both solution and solid-state NMR studies are correlation methods relying on ^1^H-^13^C, ^1^H-^15^N or ^1^H-^13^C-^15^N 2D or 3D methods. While uniform stable isotope labelling results in signal enhancement from all amino acids alongside revealing coupling networks, as with ^1^H tightly coupled systems, it may also lead to spectral overcrowding and signal degeneracy, particularly from amino acid repeats, and may hinder resonance assignment. Selective labelling by the strategic incorporation of labelled amino acids during recombinant protein expression can complement uniform methods to alleviate degeneracy and significantly simplify spectroscopic content [15]. In addition, selective labelling of amino acid CO, C_α_ CH_3_ or other carbon side chain labelling strategies can be employed to complete spectral assignment or to obtain a sufficient set of constraints [16]. A selective methyl methionine labelling scheme was demonstrated to be instrumental in elucidating the functional dynamics of β_1_AR, a GPCR, in the presence of ligands with different degrees of efficacy and G-protein mimetics, as the labelling scheme was able to reduce spectral overlap while maintaining good spectral resolution in solution NMR despite the large size of the protein complex [17]. A study of methyl methionine labelled and partially deuterated GPCR, β_2_AR, successfully probed the relative population of different protein conformation in detergent micelles and in a lipid bilayer with bound ligands [18].

## 3. Membrane Mimetics

Due to the amphipathic nature of membrane proteins, their production, purification and isolation is significantly more challenging than the analogous preparative approaches applied to soluble proteins. In addition, structural characterisation by NMR demands physical characteristics of membrane protein samples that are often contradictory to the stabilising presence of lipid membranes and the optimal preparative conditions. For example, solution NMR studies are only possible when random molecular reorientation spontaneously occurs on the GHz timescale and solid-state NMR requires very high protein to lipid ratios and optimal packing, while dynamic nuclear polarisation (DNP) signal enhancement requires optimised hydration in designer radical cocktails for optimal signal enhancement. Besides the physical state of the membrane protein systems, the surrounding lipid environment can significantly influence their structure and function. At present, structural studies of membrane proteins are not possible in their native membrane environment and structural characterisation is entirely reliant on production, isolation from exogenous membranes and purification of recombinant membrane proteins and their reconstitution into suitable, well-controlled membrane mimicking or model membrane system [19]. Some of the membrane mimetic systems are illustrated in Figure 1 and will be further described in this section.

### 3.1. Detergent Micelles

Detergents have amphipathic molecules with a large headgroup and a single aliphatic chain, which, in aqueous media above a certain critical micelle concentration (CMC), can assemble spontaneously into small, globular micellar structures. The use of detergents in studies of membrane proteins is common and often essential due to their ability to disrupt lipid membranes, solubilise membrane protein components into proteodetegent micelle complexes (PDC) and support the membrane protein fold in an aqueous, non-bilayer environment [20]. This approach is used routinely during membrane protein purification and preparation for biophysical studies, as well as in preparation for crystallisation [21]. Solubilising membrane proteins in detergent micelles is commonly used in structure determination by solution NMR, due to the comparatively small size of PDCs, which permits rapid reorientation and results in higher resolution NMR spectra. However, while detergent micelles can solubilise membrane proteins, the micellar environment is not considered a good membrane mimic, as it provides only limited support for the membrane protein fold and solubilised samples often have a limited shelf-life and aggregation is noticeable within days. In addition, the micellar composition is starkly different from the physiologically controlled and important molecular diversity of native membranes, which often affects both the structure and function of membrane proteins.

Optimal choice of detergent in membrane protein studies is essential and highly protein-specific and, despite some general considerations, is often determined by trial and error [22]. In addition, only a handful of detergents are compatible for structural studies by solution NMR due to the additional requirement for small size of detergent micelles. Three major types of detergents refer to the headgroup charge and polarity: ionic, non-ionic and zwitterionic [23]. Ionic detergents, such as sodium dodecyl sulphate (SDS), are very effective at solubilising membrane proteins but can also disrupt protein hydrophobic interactions that are essential in maintaining protein fold. This can lead to protein unfolding and denaturation, and ionic detergents are considered as “harsh”. However, SDS micelles are comparatively small in size and can lead to PDCs suitable for solution NMR studies when protein stability is not compromised [24]. Non-ionic detergents like Triton X-100 (TX-100), n-dodecyl β-d-maltoside (DDM), polyoxyethylene sorbitol ester (Tween) and n-Octyl-β-d-Glucopyranoside (OG) are considered “mild” detergents because they often affect only protein–lipid or lipid–lipid interactions. Zwitterionic detergents combine ionic and non-ionic properties. Some examples of zwitterionic detergents include lauryldimethylamine oxide (LDAO), dodecylphosphocholin (DPC or Fos-Choline-12), 1,2-dihexanoyl-sn-glycero-3-phosphocholine (C6-DHPC) and 1,2-diheptanoyl-sn-glycero-3-phosphocholine (C7-DHPC). Zwitterionic detergents are often used in solution NMR studies of membrane protein structure, partly because of their “protein-like” action and partly because of the small PDC compared to other detergents.

Once the appropriate detergent has been determined for the membrane protein under investigation, it is important to obtain an optimal concentration that provides a balance between spectral resolution and protein stability. For structural studies of membrane protein by solution NMR, the size and thermal stability of PDC need to also be carefully considered. The added complexity by this optimisation process is crucial but often extensive, costly and time consuming. Despite these challenges, detergent micelles remain a popular tool in solubilising membrane proteins for structural studies by solution NMR, as the endpoint is often a set of high resolution and high information content spectra that can yield a sufficient set of constraints and, ultimately, the protein structure in this mimetic system.

### 3.2. Bicelles

Bilayered micelles or bicelles are a mixed lipid–detergent model system for biophysical and structural characterisation of membrane proteins by solution and solid-state NMR [25]. Bicelles structurally consist of a small lipid bilayer disk, roughly 30–50 nm in size, with its hydrophobic rim stabilised by a semitoroidal detergent interface [26]. The bilayer parts of bicelles can have more complex compositions that may include other lipids, cholesterol, membrane proteins or peptides. Due to their small size, bicellar suspensions and proteobicellar mixtures are transparent and have the optical appearance of aqueous solutions, even when loaded with membrane proteins [27]. Bicelles form spontaneously from certain combinations of phospholipids and detergents and undergo a temperature-dependent phase conversion between a micellar and a bicellar phase [28]. Bicelles made of ester-linked phospholipids have a tendency to oxidise and ether-linked lipid systems are available where bicelles with longer shelf-life are needed.

The size and topology of bicelles depend on the molecular nature and the ratio of lipid to detergent, known as the *q* value, as well as on temperature. Smaller, isotropic bicelles (*q* < 0.5) are often used for solution NMR to obtain partially aligned liquid crystalline systems that restrict angular excursions of proteins in solution and permit the observation of residual dipolar couplings that aid signal assignment. Small bicelles are also able to accommodate membrane proteins within the bilayer component, which is advantageous to retaining protein fold over the use of detergent micelle mimetics.

Bicelles with a larger *q* value (*q* > 2.5) form larger discs that align magnetically with discs normally perpendicular to the magnetic field. This particular alignment results from the negative value of the anisotropic lipid diamagnetic susceptibility. The orientation of bicelles can be aligned parallel to the magnetic field by the addition of lanthanide ions [29]. This is associated with a phase conversion to stacks of perforated lamellae with detergent-capped holes [30].

The larger bicellar structures are suitable for solid-state NMR, as they provide sufficiently large lipid bilayer segments able to accommodate membrane proteins, peptides or other membrane constituents while offering orientational control for improved spectral resolution. Solid-state NMR experiments such as PISEMA (polarisation inversion spin-exchange at the magic angle) [31] were developed to probe the tilt angle of α-helical transmembrane domains with respect to membrane bilayer normal and backbone structure of protein topology. This is achieved by analysing the correlations of ^1^H-^15^N dipolar couplings and ^15^N chemical shifts [32,33].

### 3.3. Liposomes

Hydrated phospholipids can spontaneously assemble into spherical lipid bilayer structures with aqueous lumen, commonly known as lipid vesicles or liposomes. Liposomes are considered good models of biological membranes due to their low curvature and bilayer organisation. Depending on the preparation protocol, the size and organisation of liposomes can vary. Liposomes consisting of a single bilayer can be obtained with size ranging from 10 s of nm to 10 s of µm. Giant unilamellar vesicles (GUV) with diameter near 10 µm can be prepared from hydrated phospholipids by the application of low frequency electric fields. Large unilamellar vesicles (LUV) with diameter around 100–500 nm are normally prepared by extrusion through polycarbonate filters and small unilamellar vesicles (SUV) with diameter between 20 and 50 nm spontaneously form from low power bath sonication of lipid suspensions. Multilamellar vesicles (MLV) form spontaneously during hydration of lipid films or powders of lipids that have neutral curvature, such as phosphatidylcholine (PC) or phosphatidylglycerol (PG). These structures consist of concentric multilamellar stacks of bilayers, separated by roughly 14 Å thick aqueous layers, and have an overall size between 1 and 10 µm. Despite the limited resemblance of MLVs to cell membranes, they are very suitable as membrane mimics as they can accommodate membrane proteins well in the low curvature bilayer stacks, also providing high protein molarity of lipids and membrane proteins that is crucial for enhancing the signal in solid-state NMR studies.

Membrane proteins, produced recombinantly or obtained by extraction from native membranes, can be purified in detergent carriers and reconstituted into liposomes in preparation for solid-state NMR studies. The solubilising detergent is normally removed by slow or fast (aided by polystyrene BioBeads) dialysis in the presence of co-solubilising lipids that results in right side-out or mixed orientation (~50/50) of the incorporated membrane protein, respectively. The efficiency of membrane protein incorporation depends on a variety of factors such as detergent type and concentration, detergent removal rate, lipid composition and the nature of the membrane protein [34]. While liposomes are too large a system for membrane protein analysis by solution NMR, they offer a better approximation of the native membrane environment than micelles and bicelles and are the primary model system of choice in solid-state NMR studies. In addition to low bilayer curvature, liposomes offer the benefit of engineering lipid complexity by incorporation of sterols, sphingomyelin, gangliosides or other native membrane hallmarks, which are often essential for the correct folding and function of membrane proteins. Smaller, surface-active peptides or peripheral membrane proteins can be introduced from aqueous solution to preformed liposomes, while hydrophobic peptides or membrane protein TMD can be co-solubilised with the lipid mixtures in apolar organic solvents (e.g., CH_3_CN or CH_3_CL/MeOH) prior to hydration.

### 3.4. Nanodiscs

Nanodiscs are proteolipid systems containing high-density lipoprotein complex and a lipid bilayer fragment. Structurally, nanodiscs consist of two copies of amphipathic α-helical membrane scaffold protein (MSP), wrapped around and stabilising a small disc of lipid bilayer [35]. Nanodiscs have been a popular membrane mimic in recent years for the study of membrane proteins by solution NMR, due to their monodispersity, tuneable lipid composition and relatively high stability. Nanodiscs also allow membrane proteins to be studied in a detergent-free environment. The size of MSP nanodisc can be tuned by adjusting the size of the membrane scaffold protein [36]. MSP1D1 is reported to produce nanodiscs with diameters of around 9.5 nm, while MSP1D1ΔH5, which is MSP1D1 with the fifth helix deleted, produces smaller nanodiscs with diameters of around 8.4 nm [37]. The size of the MSP nanodisc is just big enough to accommodate one copy of the membrane protein embedded in the lipid bilayer but small enough to tumble rapidly and provide high resolution NMR spectra.

Membrane proteins can be assembled into nanodiscs by co-dissolving the protein, the lipid and MSP in the same detergent. Nanodiscs form spontaneously as the detergent is removed in the presence of BioBeads [38]. The stability of nanodiscs can be improved further by using circularised membrane scaffold protein (cMSP), where the N and C termini of the MSP are chemically linked together using a sortase reaction [39,40]. The chemical joining allows the MSP belt to be more resistant to heat-induced aggregation, as well as aggregation caused by elevated nanodisc concentrations [39].

Nanodiscs provide a controlled lipid environment with a homogeneous particle distribution. It has been used successfully in structural studies of membrane protein by solution NMR [41,42], including outer mitochondrial voltage-dependent anion channel (VDAC) [43,44], the TMD from bacterial outer membrane assembly factor BamA [45], outer membrane proteins from *P. aeruginosa* OprG and OprH [46] and members of the GPCR family of proteins, such as rhodopsin [47] and β_1_AR [18]. Nanodiscs have also provided insights into the effects of lipid composition on the function of membrane proteins. While the disc size and protein orientation appear unaffected by lipid composition, the proton pump function of bacteriorhodopsin appears sensitively modulated by lipid composition; both lipid headgroup charge and by the hydrophobic tail composition [48,49]. Circularised nanodiscs have also been used to extract membrane proteins from native membranes with the assistance of small amounts of mild detergent—for example, when isolating trimeric bacteriorhodopsin from the native purple membrane of *Halobacterium salinarum* [50].

### 3.5. Alternative Membrane Mimetic

A few other membrane mimetics have been used successfully to study membrane proteins by NMR. One example includes amphipols, a family of amphipathic polymers that can solubilise membrane proteins by wrapping around the hydrophobic portion of the protein [51,52] in a similar manner to detergent micelles. However, while amphipol can provide a detergent-free environment for membrane protein studies, as with detergents, amphipols mark a departure from the structure of lipid bilayers and native membranes.

Another membrane mimetic system used in NMR studies of membrane proteins is the recently developed saposin A lipoprotein discs, known as Salipro nanoparticles [53,54]. The Salipro particles contain two or more saposin A (SapA) proteins, a sphingolipid activator protein in lysosomal acidic pH [55], coalesced together and arranged in V shapes around a small disc-like lipid particle [54]. The SapA belt around the lipids is discontinuous between SapA monomers, allowing Salipro to be flexible and accommodate proteins of various sizes without strict size restrictions. Salipro nanoparticles can be assembled by mixing SapA with mild detergent, which would open up the protein conformation and permit fusion with detergent-solubilised membrane proteins and lipids. Similar to nanodiscs, upon detergent removal, the particles form spontaneously to encircle the membrane protein and lipid bilayer patch [53]. Outer membrane protein X (OmpX) from *E. coli*, a well characterised protein in detergents and nanodiscs, was successfully incorporated into Salipro and the 2D ^1^H-^15^N heteronuclear single quantum correlation (HSQC) spectrum of OmpX in Salipro was similar to OmpX in MSP1D1ΔH5 nanodiscs. A ^15^N TRACT NMR experiment showed that OmpX incorporated into Salipro forms a smaller particle (87 kDa) than OmpX incorporated into a small nanodisc (107 kDa) [53]. The use of such membrane mimetics allows membrane proteins to remain functional, as observed in reconstituted β_1_AR in Salipro [53].

Another frequently used mimetic system, engineered to support small bilayer patches in aqueous suspensions, relies on the use of block co-polymers. Similar to nanodiscs, styrene–malic acid co-polymer lipid particles (SMALPs) [56], or Lipodisqs [57], are small discotic assemblies including a lipid bilayer patch stabilised by an amphipathic organic co-polymer packing the disk rim. However, a major difference between SMALPS and MSP nanodiscs is that styrene–malic acid (SMA) co-polymer is also able to “cut-out” patches of lipid bilayer directly from the membrane without the use of detergents [56]. This mechanism relies on initial electrostatic interactions with the membranes, followed by hydrophobic insertion [58], which brings the advantage of obviating the detergent screening and solubilisation process. SMA co-polymers have been shown to extract membrane proteins from both synthetic liposomes [57] and from biological membranes [59,60,61], while preserving lipid composition and protein activity. The size of SMALPs can be tuned by adjusting the lipid–SMA polymer ratio. However, while SMALPs have the potential to produce discs either small enough for solution state NMR or big enough for solid-state NMR, their application in membrane protein structure determination by NMR has been limited [62]. Larger SMALPs have been shown to align with the magnetic field, akin to bicelles, and this alignment can also be flipped from perpendicular to parallel by the addition of lanthanide ions [63,64].

## 4. Solution NMR

Protein structure determination by solution NMR relies on obtaining a sufficient set of dihedral angle and distance constraints to calculate secondary and tertiary protein folds. A 2D ^1^H-^15^N HSQC spectrum is often recorded from ^15^N-labelled protein to obtain preliminary information, such as secondary structure and sample quality, as ^1^H-^15^N HSQC yields a single cross peak per amino acid with the exception of proline. For sequential backbone assignment, triple resonance (^1^H, ^15^N and ^13^C) 3D experiments are used in order to obtain inter and intra residue correlation information. After residue assignment, ^1^H-^1^H distance constraints from nuclear Overhauser effect (NOE), dihedral angles and hydrogen network restraints are derived from the NMR data for structure calculation.

Studying membrane protein structures by solution NMR is difficult due to limitations on sample preparation and molecular size. Membrane protein structures determined by solution NMR are restricted by the need for the protein assembly to undergo thermal, rapid rotational diffusion with correlation times below 100 ns. This limits structure determination to membrane proteins that can be embedded in micelles, small bicelles and nanodiscs. Beta-barrels, as a class of membrane proteins, are easier to characterise structurally by solution NMR than α-helical proteins due to an abundance of detectable long-range ^1^HN-^1^HN NOEs between neighbouring β-strands and due to their relative stability in these mimetics. Whilst structural analysis of helical TMD membrane proteins is complicated by low production yields of isotopically labelled protein, degeneracy in chemical shifts and comparatively few interhelical NOEs, such studies have gathered momentum with the development of techniques designed to extract interhelical structural information, such as methyl specific labelling, residual dipolar coupling measurements and paramagnetic relaxation enhancement experiments [65,66,67,68,69]. With this battery of NMR tools and membrane mimetics, a number of membrane protein structures have been resolved recently [12,62].

The human VDAC-1 structure was solved in LDAO micelles, revealing a 19-stranded β-barrel supported by inter-strand hydrogen bonding, shown in Figure 2 [70]. The assignment was achieved using three-dimensional transverse relaxation-optimised spectroscopy (TROSY) experiments at 900 MHz and the structure was calculated from over 600 NOE contacts obtained from 3D and 4D proton nuclear Overhauser enhancement spectroscopy (NOESY) experiments on a deuterated background. VDAC-1 was the first eukaryotic barrel solved in solution, with the protein being the most abundant in mitochondrial outer membranes. The biological role of VDAC-1 involves transporting solutes and ions across the membrane as well as playing a key role in mitochondria-induced apoptosis.

Bacterial outer membrane protein X (OmpX) from *E. coli* was another β-barrel membrane protein structure solved early by solution NMR. Its structure has been determined in DHPC [71,72] and DPC [37] micelles, as well as in phospholipid nanodiscs [37,73]. Using different mimetic environments resulted in differences between the final structures predominantly within the extracellular loops, chiefly their length and orientation with respect to the membrane. This highlights the important role of choosing the membrane protein support system, as using different membrane mimetics affects membrane protein conformation as well as function. The structure of outer membrane protein A (OmpA) from *E. coli* was solved in DPC micelles using a number of TROSY-based experiments and a combination of 600 and 700 MHz spectrometers to match the protein relaxation characteristics during backbone assignment. The final structure was based on 91 NOE distance constraints, 142 dihedral angle constraints and 58 hydrogen bond restraints that allowed the authors to calculate the β-barrel fold of OmpA [74]. The total structure of *E. coli* outer membrane protein G (OmpG) was solved as the largest monomeric membrane protein at the time with 234 of 280 residues assigned using TROSY-based experiments on an 800 MHz spectrometer and 133 long-range NOEs were used to calculate the 14-strand β-barrel global fold [75].

Phototaxis receptor sensory rhodopsin II (pSRII) was the first seven-helix transmembrane protein to be structurally characterised by solution NMR in DHPC micelles [76]. pSRII is involved in blue-light avoidance by *Natronomonas pharaonis.* The protein structure was solved using chemical shift-derived dihedral angles, hydrogen bonds and long-range methyl NOEs. The structure was in good agreement with the known crystal structure [76]. Solution NMR also solved the structure of the 5-transmembrane α-helical mitochondrial translocator protein (TPSO). TPSO was solubilised in DPC micelles and with the aid of perdeuteration and methyl labelling, and 98% and 95% assignment was achieved of backbone and side chain resonances, respectively [77]. This generated over 3300 NOE distance restraints that were used for structure calculations.

Solution NMR was also used to determine the structure of another helical membrane protein, the p7 cation channel from hepatitis C. Chou and colleagues determined a hexameric assembly of the p7 protein in DPC micelles, in which the protein formed a funnel-like structure with a flower shape [78]. However, this structural model has been disputed by Zitzman, Schell and colleagues, who have presented evidence that in DPC micelles, the protein is monomeric [79,80]. The latter investigation used a higher excess of DPC than Chou et al. [81], who subsequently showed that p7 does form a hexameric structure in isotropic bicelles. This highlights the importance of detergent to protein ratio as it has been shown that too high a ratio can cause the protein to denature and misfold or alter protein–protein and other biological interactions [82].

A comparison between solid-state and solution NMR structural data can offer further insights into the role of mimetic environments used to solubilise membrane proteins, both in terms of structure and dynamics. An illustration of such effects using OmpX embedded in nanodiscs revealed that following measurements using solution and solid-state NMR, protein structure and dynamics were affected above and below the lipid transition temperature [83].

## 5. Solid-State NMR

Solid-state NMR is well suited to structural studies of membrane proteins, as it is capable of interrogating protein conformations in their native or close to native lipid environments. Despite this advantage, it has not found wide use in the total structural determination of membrane proteins due to challenges with residual inhomogeneous line broadening. Instead, solid-state NMR has been developed and used extensively as a sensitive tool in probing specific molecular interactions and structural dynamics. Solid-state NMR can also be used in combination with solution NMR for detailed characterisation and chemical shift assignments. Unlike solution NMR, anisotropic nuclear interactions are managed externally and free molecular mobility within the sample is not required in high resolution solid-state NMR spectroscopy. Orientation-dependent anisotropic interactions that normally lead to spectral broadening in solution NMR, such as chemical shift anisotropies and dipolar couplings, dominate the spectroscopic features in the solid state. These interactions carry a wealth of orientational information and can be used to report on molecular structure or interactions or can be removed to obtain high resolution spectral information. Anisotropic line broadening in solid-state NMR is usually mitigated by one of two different approaches, either using oriented samples (OS) or by magic angle sample spinning (MAS).

Sample alignment can be generated for membrane protein studies either by mechanically orienting samples on physical supports, such as glass plates [84], or by using self-aligning membrane mimetics, such as magnetically aligned bicelles [27]. Large SMALPs have also been shown to align magnetically, much like bicelles [63], and may offer a detergent-free system of OS-NMR studies of membrane proteins in lipid bilayers. OS-NMR offers insights into the tensor orientation of molecular phosphate effective chemical shift anisotropy (CSA), revealing the impact of membrane protein insertion in membranes. Two-dimensional PISEMA correlation spectra using ^15^N chemical shifts and ^15^N-^1^H dipolar couplings from ^15^N labelled α-helical peptides and proteins reveal an elliptical pattern, known as the polarity index slant angle (PISA) wheel, which is dependent on the tilt angle between protein helical domain and the membrane [32,33]. This approach has been successful in characterising membrane proteins and membrane-associated peptides, such as the M2 transmembrane peptide from the nicotinic acetylcholine receptor (nAChR) [85] and CXCR1 from the GPCR family of proteins [86]. Almost all transmembrane helices of membrane proteins are found tilted with respect to the lipid bilayer normal [87] and the helical ends are often anchored to the membrane surface by interfacial tryptophan residues [88].

In an alternative approach, aimed at obtaining high resolution NMR spectra for structural studies of membrane proteins, the anisotropic interactions are removed and subsequently selectively reintroduced to obtain distance and dihedral angle constraints. Orientation-dependent second order tensor interactions that dominate solid-state lineshapes scale with the rotation axis tilt angle as $P_{2}\left(cos\theta \right)=\frac{1}{2}\left(3cos^{2}\theta -1\right)$ and can be removed by macroscopic sample rotation at speeds exceeding the strength of these interactions about the “magic angle”, where $P_{2}\left(cos\theta \right)$ vanishes. The “magic angle” of 54.7° sets the alignment of the rotation axis along the crystallographic [111] direction and the external magnetic field at [001]. MAS is usually combined with electromagnetic pulsed radiofrequency-driven nuclear nutation, which permits both the removal and subsequent selective reintroduction of nuclear couplings that provide distance and dihedral constraints. This results in high resolution NMR spectra dominated by scalar isotropic chemical shifts and J-couplings from membrane proteins and other membrane constituents.

The strength of some nuclear interactions, such as homonuclear ^1^H-^1^H dipolar couplings that are common in proteins, exceeds the routinely accessible MAS speeds of 10 s of kHz. As a result, proton detection that is almost universal in biomolecular solution NMR is often out of reach in solid-state measurements, as residual couplings broaden proton lines and reduce spectral resolution and ^1^H spectroscopic content. This can be addressed at moderate spinning speed (20–50 kHz) by isotope dilution, in which membrane proteins can be studied in proton-detected experiments either after full [89] or partial proton back-exchange [90]. In addition, recent developments in ultrafast MAS instrumentation have opened the possibility of proton detection and high resolution solid-state NMR from protonated membrane protein samples. Ultrafast MAS (>60 kHz and above 100 kHz) can lead to complete averaging of homonuclear ^1^H couplings and well-resolved two dimensional ^1^H-^15^N spectra can be recorded from fully back-exchanged membrane proteins embedded in lipid bilayers [83,91] or form protonated crystalline protein samples [92].

The most commonly observed nuclear system in solid-state NMR structural studies of membrane proteins is ^13^C, due to its large chemical shift dispersion in biological macromolecules and comparatively small linewidth. To obtain structural and geometric constraints, proteins are enriched in ^13^C (often in combination with ^15^N) and studied using two-dimensional homonuclear (^13^C-^13^C) or heteronuclear (^13^C-^15^N) correlation spectroscopy. A wealth of experimental tools and pulse sequences has been developed to probe molecular interaction networks and extract structural information exploring nuclear interaction either through-space (dipolar couplings) or through-bond (scalar couplings). Commonly used sequences for obtaining homonuclear correlations, such as proton-driven spin diffusion (PDSD) [93], dipolar-assisted rotational resonance (DARR) [94] and radio frequency-driven recoupling (RFDR) [95], provide chemical shift assignment and distance information that is used for structure determination. Carbon–carbon homonuclear correlation spectroscopy has been used to determine membrane protein structure in membrane [96,97,98,99], membrane protein conformational changes [100] and to interrogate molecular interactions within multicomponent membrane protein complexes [101].

In addition to homonuclear single-quantum correlation spectroscopy, double-quantum (DQ) methods have been developed to observed specific spin interactions. Double (or higher) quantum excitation can only be observed in dipolar-coupled systems if a neighbouring spin is proximal in space and within a rigid environment. DQ excitation methods include POST-C7 [102], R symmetry-based R14 [103] and five-fold symmetry-based SPC5 [104], all of which rely on scrambling second order nuclear interactions and selectively reintroducing the desired dipolar couplings. Double quantum excitation can also be used as a filter (DQF), which removes long intramolecular interactions and undesirable background signals [105], particularly useful in carefully designed selective ^13^C labelling schemes and high natural abundance ^13^C background. Homonuclear ^13^C-^13^C correlation spectroscopy with DQF can also be used to study protein–lipid interactions, such as identifying cholesterol interaction with ^13^C labelled transmembrane helix of influenza M2 protein in model membranes [106]. Double quantum–single quantum (DQSQ) correlation spectroscopy is used to observe coherence evolution within the DQ state as a sensitive way of deriving distance-dependent couplings and constraints. DQSQ correlation spectroscopy demonstrated further improvement of spectral resolution and suppression of background signals, as well as a good chemical shift separation difference, and is useful in studies of larger or more complex systems. DQSQ ^13^C-^13^C correlation spectroscopy has been used to study membrane protein structure while removing background lipid signals and providing backbone chemical shift assignment [107].

One-dimensional solid-state NMR techniques can also be applied to seek nuclear coupling membrane protein structures and conduct interaction studies. Rotational echo double resonance (REDOR) [108] is a recoupling sequence that reintroduces distance-dependent heteronuclear dipolar couplings to obtain accurate distance information with sub-angstrom resolution. In combination with site-specific isotope incorporation, REDOR can be used to study peptide–membrane and protein–membrane interactions, identifying membrane localisation of specific residues in membrane-associated proteins [109], as well as conformational changes in membrane proteins with bound ligands [110].

Reporter nuclei within biological macromolecules are considered abundant (^1^H) or rare (^13^C, ^15^N) because of their natural abundance, relative interconnectivity and individual degree of polarisation, all of which affect the magnitude of the observed signal. Cross polarisation (CP) is the most commonly used method for indirect excitation of rare nuclei and signal enhancement in solid-state MAS NMR [111], often deployed as a first step in complex multipulse and multidimensional experiments. CP relies on polarisation transfer from an abundant nuclear system with high γ, such as ^1^H, through strong dipolar coupling to the rare spin via Hartmann-Hahn contact [112]. Signal enhancement is achieved due to transfer of the stronger proton polarisation onto the rare spin system (such as ^13^C or ^15^N), and to a reduction in recycle delay between experiments. The efficiency of polarisation transfer crucially relies on heteronuclear dipolar coupling from the proton to the directly attached rare nuclear system, which is only possible in motionally restricted molecular systems. In membrane studies under physiological conditions, CP can be used as a motion filter to select only signals arising from relatively rigid molecular subsystems and CP can be used to probe lipid–protein interaction of peripheral or membrane-associated proteins [113]. Shown in Figure 3, the interaction between the docking site of prion protein and membrane-embedded ganglioside GM1 leads to attenuation of signals from contact sites [113].

Solid-state NMR is uniquely suited to studies of membrane protein systems in larger assemblies and multiprotein membrane complexes within synthetic or native cell membranes. For example, the β-barrel assembly machinery (BAM) complex from *E. coli* includes five protein partners, which coordinate the integration of β-barrel outer membrane proteins into the outer membranes of Gram-negative bacteria, mitochondria and chloroplasts. BamA, the protein hub, has been studied using a combination of specific isotopic labelling schemes with solution and solid-state NMR [114]. Solid-state NMR was possible within the fully assembled complex in native *E. coli* outer membranes, confirming the overall fold and assembly of the Bam complex [101].

Voltage-gated potassium channels, KcsA, monitor potassium ion transport through the membrane and are essential for the regulation of neuronal and cardiac action potentials. Solid-state NMR has been an important tool in the study of potassium channels, providing insight into membrane-embedded protein structure, dynamics and informing on the functional conformation of the protein [98,115,116,117,118]. Solid-state NMR studies have shown that changes in membrane lipid composition alter the activation state of KcsA and the presence of cardiolipin increases the stability of the open-conductive conformation of the channel [100]. CXCR1, a member of the GPCR family of proteins shown in Figure 4, has been characterised using solid-state NMR using both OS-NMR and MAS, providing a three-dimensional structure in membranes under physiological conditions, and studies have shown the importance of phospholipid bilayer structure and composition for maintaining the functional state of the protein [86,97,119].

## 6. Dynamic Nuclear Polarisation NMR

One problem, common to both solution and solid-state NMR, is their relatively low sensitivity, which arises from the low nuclear polarisation and which has necessitated the use of large protein amounts and high sample concentrations in almost all membrane protein studies. As a result, NMR studies of native membrane systems and whole cells are few and far between. One way of enhancing NMR signal intensity relies on the use of paramagnetic radical systems, coupled to the nuclear system of interest, in combination with dynamic nuclear polarisation (DNP).

DNP was described in the late 20th century but has recently emerged as a powerful approach to tackling the problem of low sensitivity from biomolecular samples with the availability of stable and reliable high-power microwave sources. In solid-state NMR experiments, low nuclear sensitivity and often low amounts of biological samples require excessively long experimental times, which limit the scope and application of the technique. Nuclear signal enhancement is achieved in DNP by exciting paramagnetic electrons in bespoke engineered biradical systems, which polarise 1000-fold better than nuclei, and transferring their polarisation to the surrounding nuclear systems [120]. The polarisation transfer is facilitated by continuous microwave (MW) irradiation at 100 K to optimise transfer and relaxation and to ensure saturation of the electron–nuclear transition. However, signal enhancement gain in DNP comes at a tradeoff in resolution as a result of the mobility restriction at low temperatures and the proximity to paramagnetic relaxation sinks. As such, DNP enhancement efficiency and spectroscopy are highly dependent on sample preparation, including sample properties and structure, preparation protocols, sample quality and monodispersity, as well as the choice of biradical. Some biradicals, commonly used in biomolecular DNP NMR, include hydrophillic TOTAPOL [120] and AMUPol [121], as well as apolar TEKPol [122]. Studies with radicals specifically tailored to the experiment have also been reported and include membrane proteins with tethered spin labels [123] and lipid anchored radicals [124]. As DNP MAS NMR experiments are performed at low temperatures, a cryoprotectant is required to ensure sample mixing and homogeneity and to prevent the formation of water crystals. This is normally done using a cryoprotectant cocktail solution, consisting of ^12^C-*d*_6_-gylcerol, D_2_O and H_2_O at a molar ratio of around 60:30:10. The composition of cryoprotectant cocktail, biradical concentration and sample of interest usually needs to be optimised for achieving maximum enhancement factors and the D_2_O and H_2_O ratio tuned for sufficient polarisation transfer and moderate radical relaxation.

The sensitivity enhancement in a DNP experiment, ε, can be measured as the ratio of spectral intensity acquired with or without MW irradiation. DNP MAS NMR has been used to study numerous membrane protein systems, and in crystalline membrane proteins bacteriorhodopsin [125,126] and channelrhodopsin [127,128], significant enhancement of ε = 40~70 has been reported, perhaps due to molecular organisation in ordered 2D crystalline arrays or protein oligomers. Membrane proteins reconstituted in liposome such as KvcA [129], EmrE [130], ATP-binding cassette (ABC) transporters [131] and GPCR B_1_R [132] have been studied using DNP MAS NMR to investigate protein conformations or protein–ligand interactions.

DNP signal enhancement and strategic localisation of the excitation biradicals provide unique new opportunities to study more challenging and complex biological systems that are otherwise intractable by solid-state NMR. Membrane patches, isolated from *E. coli* culture containing membrane protein Mistic, have been characterised [133] and Bcl-XL from the Bcl family found in mitochondria was studied from cell lysate with biradical-labelled ligand [134]. Natural membrane extracts of nAChR bound to toxin and reconstituted in membrane patches of *T. californica* showed enhancement ε = 12, allowing 2D ^13^C correlation experiments to be recorded in 14 h instead of 9 days [135]. DNP NMR studies of whole bacterial cells, both Gram-positive *B. subtilis* [136] and Gram-negative *E. coli* [137], as well as human embryonic kidney cells [138], have been reported, showing the great promise and potential of DNP MAS NMR as a tool for studying membrane proteins in native membrane environments and cells.

## 7. Conclusions

A revolution in structural biology over the past 30 years has left behind a vast and crucially important field of membrane protein structure determination. Recent advancements in membrane protein X-ray crystallography and cryo-TEM have shown great promise and have provided a number of template structures that have inspired a boom in homology modelling and MD simulations of membranes and membrane proteins. At the same time, NMR remains the leading tool for structural and dynamics studies of membrane proteins, protein complexes, as well as protein–ligand and protein–lipid interactions, under physiological conditions. Superconductive NMR magnets have exceeded the 1 GHz mark and low noise cryo-probes offer superb low noise spectral acquisition. Ultrafast MAS probes reaching speeds in excess of 100 kHz are now widely available alongside robust, fast-switching, high-power solid-state consoles. System complexity and departure from the confinement of the crystalline environment have yielded a range of membrane mimetics, which have enabled longer and higher resolution measurements from more complex and more realistic membrane systems. The availability of medium and high field solid-state NMR instruments has delivered structures from native membranes, and early work in cells is emerging. Despite this success, NMR remains confined by low sensitivity and a need for large and concentrated membrane protein samples, and routine whole cell work is just out of reach. With recent commercial development of high power microwave generators, DNP has come to the fore to deliver 10 to 100 s of time signal enhancement, carrying the promise of localised and selective signal enhancement for membrane proteins in native membranes and in whole cells.

## Figures and Tables

**Figure 1 biology-09-00396-f001:**
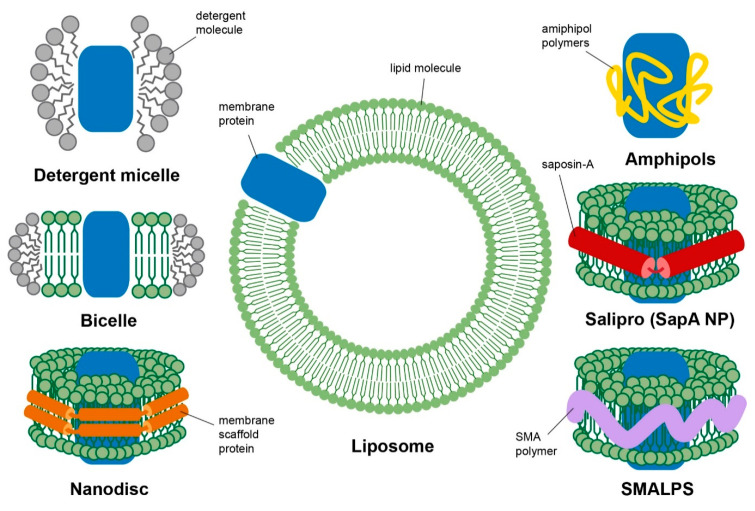
Schematic diagram of different membrane mimetic systems used in NMR studies: detergent micelle, bicelle, nanodisc, liposome, amphipols, salipro and styrene–malic acid co-polymer lipid particles (SMALPS). Membrane protein is represented as a blue cartoon block.

**Figure 2 biology-09-00396-f002:**
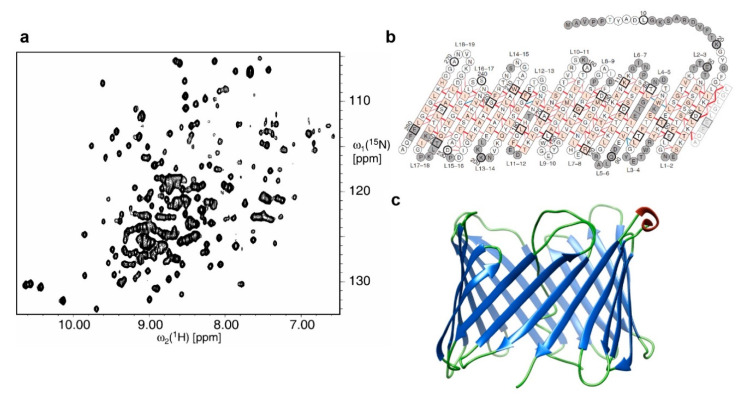
Solution state NMR study of human VDAC-1 in LDAO micelle: (**a**) two dimensional ^15^N-^1^H HSQC spectrum of uniformly labelled ^15^N, ^2^H human VDAC-1 detergent micelles, (**b**) amino acid sequence of VDAC-1 arranged in secondary and tertiary structure and (**c**) cartoon representation of the three dimensional solution NMR structure, where β sheets are shown in blue, α helices in red and coils shown in green [PDB:2K4T]. Figure adapted with permission from [70], copyright 2008 American Association for the Advancement of Science.

**Figure 3 biology-09-00396-f003:**
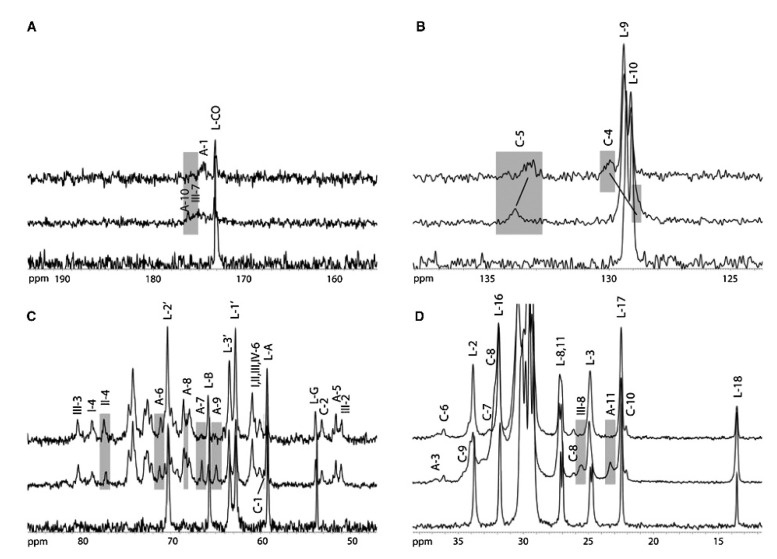
Carbon-13 CP MAS NMR spectra from membranes containing raft marker, ganglioside GM1; specific resonance attenuation reveals the binding epitope for prion proteins on GM1. The spectra show chemical shift assignments and highlight changes between (top spectrum) POPC/GM1/PrP, (middle spectrum) POPC/GM1 and (bottom spectrum) POPC, in (**A**) the carbonyl region, (**B**) the HC=CH region, (**C**) the oligosaccharide region and (**D**) the hydrocarbon chain and N-acetyl methyl groups. Reproduced with permission from [113], copyright 2011 Elsevier.

**Figure 4 biology-09-00396-f004:**
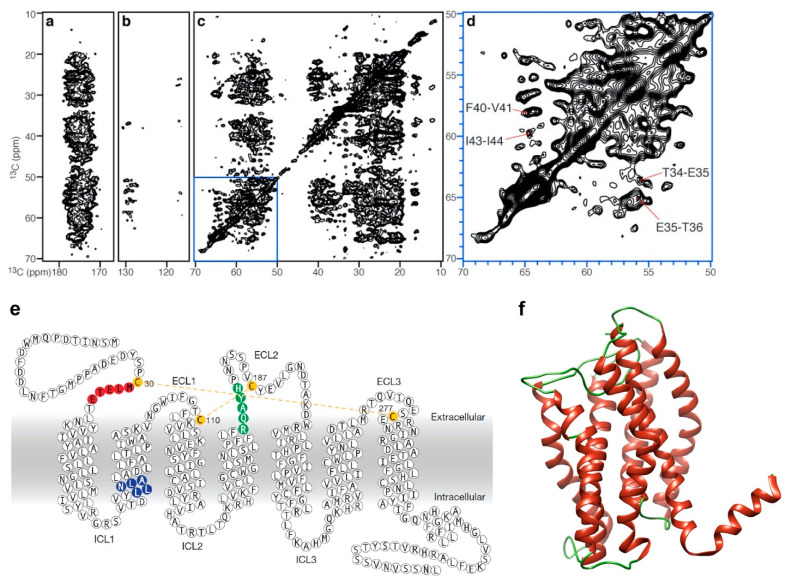
Solid-state NMR study of human CXCR1 in lipid bilayer: (**a**–**d**) two dimensional MAS ^13^C-^13^C PDSD spectrum of uniformly labelled ^13^C CXCR1 in liposome showing specific regions of (**a**) carbonyl carbons, (**b**) aromatic carbons, (**c**) aliphatic carbons and (**d**) is the expanded spectral region highlighted by the blue box; (**e**) amino acid sequence topology of CXCR1 and (**f**) cartoon representation of the three dimensional solid-state NMR structure, where α helices are shown in red and coils shown in green [PDB:2LNL]. Figure adapted with permission from [97], copyright 2012 Springer Nature Limited.

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
