# Peer review of "Membrane Protein Structure Determination and Characterisation by Solution and Solid-State NMR"

_biology, 2020, doi:10.3390/biology9110396_

Round 1

Reviewer 1 Report

In this contribution Yeh et al provide a comprehensive overview of the use of solution and solid state NMR techniques for the characterization of membrane proteins. They cover several aspects of sample preparation (including isotopic labeling of the protein and choice of the optimal membrane mimetic system), introduce conventional and advanced NMR methods for structural characterization of membrane proteins, and describe a series of successful examples in which NMR was employed to characterize the structure/function relationship in biologically relevant systems.

Overall I think that this contribution is important as it introduces complex and advanced biophysical experiments using a simple and accessible language that avoids the excessive use of mathematical formulas. In my opinion, these contributions complement the plethora of highly NMR specialized reviews available in the literature and will help spreading to the broader scientific community the message that several NMR methods are available for structural characterization of membrane proteins. In light of this comment, I think the authors should consider citing https://doi.org/10.3389/fmolb.2020.00009 when they introduce the versatility of NMR in reporting on intermolecular interactions (page 2). The latter minireview describes conventional solution and solid-state NMR experiments for studying protein-protein interactions in a simple and math-free language.

Author Response

We thank the reviewer for the thorough read and for their constructive comments. All changes have been made, as requested and the reference has been added.

Reviewer 2 Report

This paper provides a review of relatively recent developments of applying solution and solid-state NMR methods to understand the structure and dynamics of membrane proteins. Importantly this includes the recent addition of dynamic nuclear polarisation techniques. Membrane proteins are a very important class of molecules. The use of X-ray diffraction and cryo-TEM are key methods for their structure determination, but membrane proteins can be difficult to prepare, for example in their crystallisation. NMR is therefore an important component technique to analyse these molecules. However there can be limitations of the NMR and these are openly discussed here. In preparing samples mimetic systems are often used and there is a very good description of such systems, which will be helpful to researchers coming to this field for the first time. The complementarity of solution and solid-state NMR approaches is described. A review article should balance its accessibility to the target reader, detail and the comprehensiveness of the coverage. This is quite well achieved here as there is a good use of references which balances up to date examples with some of the foundational papers. The writing is OK. In places it is not always a model of clarity, but most of the time it is sufficiently clear. It could helpfully do with a further careful edit. There is one mistake that must be corrected (1, below) and a number of others that are simple typographical/constructional errors that need to be tidied up. This is not a comprehensive list, but illustrates the errors that should be looked for. This is a helpful addition to the literature and once minor corrections and editing by the authors have been done this will be acceptable for publication.

Minor error which must be corrected.

  1. p11, last paragraph, line 5, it is not cosθ2, it is cos2θ

Minor errors for tidying

  1. Throughout out there are several inconsistencies such as solid state/solid-state, polarisation/polarization, etc. I have no strong preference for any of these, but they must be consistently referred to throughout.

  1. p3, line 2, analysis of should be analyse

  1. p4, line 20, PGCR should be GPCR

  1. p5, line 6, and should be an

  1. p6, Sec. 3.2, line 5m cholesterol is mis-spelt

  1. p6, penultimate line, is should follow This

  1. p7, line 5, not clear what 10ns stands for

  1. p7, line 13, tick should be thick

  1. p8, Sec. 3.5, line 17, characterised is mis-spelt

  1. p8, Sec. 3.5, line 20, incorporated is mis-spelt

  1. p9, Sec. 4, line 8, should be restraints

  1. p9, Sec. 4, line 15, should be strands

  1. p10, line 3, should be well

  1. p10, line 9, there should be a space to give 700 MHz

  1. p11, Sec. 5, line 20, should be insights

  1. p11, Sec. 5, line 22, should be membranes and PISEMA

  1. p12, line 2, should be from

  1. p13, line 11, stronger should replace better

  1. p13, line 12, a should be inserted before reduction

  1. p14, line 7, or should be of

  1. p14, Sec. 6, line 2, it is the polarisation not the polarizability

  1. p15, line 10, delete the is after DNP

  1. p16, line 7, should be structure

Author Response

We thank the reviewer for the thorough read and for their constructuve comments. All changes made in the revised manuscript, as requested.